# Mechanisms for the prevention of adolescent intimate partner violence: A realist review of interventions in low- and middle-income countries

Hattie Lowe[1]*, Joanna Dobbin[2], Ligia Kiss[1], Joelle Mak[3], Jenevieve Mannell[1], Daniella Watson[4], Delanjathan Devakumar[1]

1 Institute for Global Health, University College London, London, United Kingdom, 2 Primary Care and Population Health, University College London, London, United Kingdom, 3 Department of Global Health and Development, London School of Hygiene and Tropical Medicine, London, United Kingdom, 4 Global Health Research Institute, School of Human Development and Health, Faculty of Medicine, University of Southampton, Southampton, United Kingdom

* hattie.lowe@ucl.ac.uk

**Data Availability Statement:** Data supporting the results presented in this article are drawn from

## Abstract

Adolescent girls are among those at the greatest risk of experiencing intimate partner violence (IPV). Despite adolescence being widely regarded as a window of opportunity to influence attitudes and behaviours related to gender equality, evidence on what works to prevent IPV at this critical stage is limited outside of high-income, school-based settings. Even less is understood about the mechanisms of change in these interventions. We conducted a realist review of primary prevention interventions for adolescent IPV in low- and middle-income countries (LMICs) to synthesise evidence on how they work, for whom, and under which circumstances. The review took place in four iterative stages: 1) exploratory scoping, 2) developing initial programme theory, 3) systematic database search, screening and extraction, and 4) purposive searching and refinement of programme theory. We identified eleven adolescent IPV prevention interventions in LMICs, most of which demonstrated a positive impact on IPV experience and/or perpetration (n = 10). Most interventions (n = 9) implemented school- or community-based interactive peer-group education to transform attitudes and norms around gender and relationships for behaviour change. The central mechanism of change related to gender transformative content prompting adolescents to critically reflect on their attitudes and relationships, leading to a reconceptualisation of their values and beliefs. This central mechanism was supported by two secondary implementation mechanisms: 1) the design and delivery of interventions: interactive, age-appropriate education delivered in peer-groups provided adolescents a safe space to engage with content and build communication skills, and 2) the target group: social norms interventions targeting the wider community created enabling environments supportive of individual change. This review highlights the immense potential of gender transformative interventions during the critical period of adolescence for IPV prevention. Future interventions should consider the broader drivers of

open access secondary published data. See S4 Table.

**Funding:** This study was funded by the National Institute for Heath and Care Research, grant number 17/63/47. The funders had no role in study design, data collection and analysis, decision to publish, or preparation of the manuscript. https://www.nihr.ac.uk/.

**Competing interests:** The authors have declared that no competing interests exist.

adolescent IPV and ensure intersectionality informed approaches to maximise their potential to capitalise on this window of opportunity.

## Introduction

Intimate partner violence (IPV) affects approximately one in three women globally, although with wide variation within and between countries [1], and has detrimental consequences for physical and mental health [2]. Adolescent girls are among those at greatest risk of experiencing IPV. Available estimates suggest that 24% of ever-partnered girls aged 15–19 years have experienced physical and/or sexual IPV in their lifetime [2]. Experiencing IPV during adolescence has been linked with poor health and social outcomes, including mental health problems, school drop-out, substance use, risky sexual behaviours, injury, and death, with long-term implications for future health and wellbeing [3].

Adolescence, defined by the World Health Organization as the period from 10–19 years of age [4], is a period of immense change. While undergoing cognitive development and major physiological changes as a result of puberty, adolescents are also faced with the huge challenge of developing a sense of self and navigating the shifts in social roles and expectations placed upon them [5]. For adolescent girls, vulnerabilities that come with physical, sexual, cognitive and social development are further compounded by the fact that they are female. Gender discrimination, norms, and practices place girls at increased risk of experiencing different forms of gender-based violence, such as sexual violence and harmful practices [6]. At the intersection of violence against children and violence against women, adolescence is a period in which IPV victimisation often begins for girls [7, 8]. This is highlighted by the fact that the global prevalence of IPV among adolescent girls (24%) closely mirrors the prevalence of IPV among all women aged 15–49 years (27%) [2, 9]. Gender inequitable attitudes and norms also disadvantage boys during adolescence. Hegemonic masculine norms which promote male dominance and toughness place boys at increased risk of substance abuse, violence experience and perpetration (including community-based violence), delinquency and other risky health behaviours such as unsafe sexual practices [10–12]. Adolescence is also widely regarded as an important life stage in which attitudes and behaviours related to gender inequality and violence can be influenced, creating a window of opportunity for intervening to establish the foundations for current and future relationships free from violence [13, 14].

The evidence base for how to prevent adolescent IPV is slowly growing, with the majority of evidence coming from high-income countries (HICs), particularly the USA and Canada. There have been relatively fewer rigorous evaluations in low- and middle-income country (LMIC) contexts, despite the burden of adolescent IPV being high [2]. In HICs, school-based prevention programmes have demonstrated promising results, with participatory and curriculum-based group learning, peer mentor training, bystander interventions and relationship skills approaches exhibiting a positive impact on gender-equitable norms, the intention of bystanders to intervene, and the experience and perpetration of adolescent IPV [9, 15–18]. Programmes in LMICs, where conceptualisations of adolescence and intimate relationships may differ (though with great heterogeneity between countries), have expanded on these approaches, with a greater focus on the gendered experience of IPV for adolescent girls, generally looking at adolescent girls as victims or survivors, and adolescent boys as perpetrators, based on theorisations from prior research [9, 17].

Despite interventions in LMIC settings showing promising results, theorisations of the mechanisms within them that bring about change are lacking [9, 19]. This limits our

understanding of how these interventions might work across different contexts, with different sub-groups, and whether they can and should be scaled-up and transferred across settings. In light of this, we use realist synthesis as the method for this review because of its theory driven principles which seek to determine how change is triggered by an intervention and under what conditions. We aim to build on previous systematic reviews of adolescent IPV prevention interventions to generate evidence on the mechanisms of change within them. We will build theory that future intervention development and evaluation can build upon for more effective and context-specific IPV prevention programmes, capitalising on this window of opportunity for change during the adolescent years.

## Methods

Realist evaluation is centred around context-mechanism-outcome (CMO) configurations which explain the pathways from an input to an output when a programme is implemented within a social system [20]. In this review, we define context as the prevailing socio-cultural, economic and political conditions in which programmes are implemented [21]. Contextual factors are critical to programme success or failure because they provide the conditions in which mechanisms are triggered or not [22]. Mechanisms, disaggregated into resources and reasoning, provide explanations of what is happening within the system to produce the observed outcomes [23]. Specifically, resources refer to the components that are introduced by the programme under study, and reasoning refers to human responses triggered by the intro-duction of the resource in that given context. Finally, outcomes deal with the intended (or unintended) changes that are brought about by the introduction of the programme within a given context. In short, we expect that when an intervention is introduced in a given context, it triggers a change in reasoning within the participants, altering their behaviour and leading to an outcome [23, 24]. The protocol for this realist review is registered in PROSPERO (refer-ence: CRD42021284160) and we followed the RAMESES publication standards for realist syn-thesis (see S2 Table) [25]. Conducting this review was an iterative and collaborative process, broadly taking place in four stages [26].

### Stage 1 –Exploratory scoping and definition of key concepts and criteria

First, we conducted a rapid literature assessment to scope the existing literature on primary prevention interventions for adolescent IPV. This involved searching for systematic and narra-tive reviews in the databases Medline, Embase, Psycinfo, Web of Science and CINAHL Plus. The inclusion criteria for this initial search was broad, focusing on reviews that synthesised evidence on primary prevention interventions for any form of IPV among adolescents in LMICs, with no restrictions on date, methods or language. From this, we were able to develop a sense of the size of the evidence base and review the existing literature to refine the scope and definitions for our own review. The interventions described in these reviews formed the basis of our initial list of included interventions.

In this review, we defined adolescent IPV according to the WHO as "behaviour by an inti-mate partner or ex-partner that causes physical, sexual or psychological harm, including physi-cal aggression, sexual coercion, psychological abuse and controlling behaviours" [27], but specifically occurring between two adolescents, aged 10–19, in an intimate, romantic or dating relationship. While there is conceptual overlap, we did not include non-partner sexual vio-lence, child abuse, sexual violence in conflict or child marriage in our definition of adolescent IPV. We acknowledge that each of these types of violence have their own body of literature and evidence on interventions and we have chosen to focus specifically on violence that occurs between two adolescents in an intimate, romantic or dating relationship. We also acknowledge

that relationships during adolescence are defined and conceptualised differently across the very different settings included in this review. While we use a more generalised definition of IPV, we do not seek to imply that adolescent relationships and IPV is the same across all LMICs. Using a realist approach enables us to explore each intervention setting in substantial depth, while still creating a generalised theory that has relevance for intervention development across settings.

We defined primary prevention interventions as any intervention that aims to prevent adolescent IPV before it occurs, for example, by preventing exposure to certain risk factors or altering unsafe attitudes or behaviours. Interventions in this review may be provided by public, private or charitable organisations and at either an individual, group, community or societal level, providing they target the adolescent population (10–19 years). Where study populations were defined as young people or youth, studies were included if either the results were disaggregated by age, or if over 50% of the study population were aged 10–19 years [9]. To be eligible for inclusion, studies had to assess, qualitatively or quantitatively, the impact of the intervention on the perpetration or experience of adolescent IPV. Study participants included adolescent survivors or perpetrators of IPV and the general adolescent population, as well as key stakeholders involved in providing the intervention. Materials that did not attempt to provide evidence for the link between the intervention and the outcome of interest (perpetration or experience of adolescent IPV) were excluded.

## Stage 2 –Development of initial programme theories

In the second stage, we built an initial theoretical model for how primary prevention interventions for adolescent IPV work, based on the materials retrieved in stage one. We started by mapping the different types of primary prevention interventions presented across the materials. We then built initial context-mechanism-outcome configurations, linking intervention types to their primary and intermediary outcomes by reviewing the theories of change, hypotheses and discussions of findings presented across the materials to suggest possible pathways (mechanisms) that explained how the different intervention modalities resulted in a reduction in the perpetration or experience of adolescent IPV. We also examined data pertaining to successful design and delivery elements of interventions [28], based on stakeholder and author reflections. Intervention protocols and the broader literature on adolescent IPV (such as cross-sectional studies on risk factors and qualitative studies of experiences) were also helpful at this stage to map out different contexts to be examined at a later stage. This initial programme theory was used to inform the systematic literature search and data extraction in stage three.

## Stage 3 –Systematic literature search for empirical evidence, screening, and data extraction

In the third stage of this review, we conducted a systematic literature search to identify any additional empirical evidence that was eligible for inclusion in our review in order to test and refine the initial programme theory. We searched the databases Medline, Embase, PsycInfo, Web of Science and CINAHL Plus. The search strategy was based around four central concepts: 1) adolescence (including all terms pertaining to people aged 10–19 years old), 2) IPV (including all terms pertaining to violence that occurs in an intimate or dating relationship or from a current or former partner), 3) LMICs (including all countries classified by the World Bank as low- or middle-income), and 4) prevention interventions (including all terms pertaining to primary prevention initiatives). Articles were screened by one author (HL). The search strategy is presented in full in Supporting Information 1 (S1 Table). To complement these

searches, we hand searched relevant journals, conducted citation chaining and contacted authors to ensure a comprehensive collation of materials. In order to retrieve materials beyond academic journal articles, we also hand searched relevant data repositories including the WHO reproductive health library, UNFPA regional webpages, World Bank Open Knowledge Repository, WHO Prevent Violence Evidence Base and Resources, WHO Institutional Repository for Information Sharing, UNDP, UN Women, WHO Reproductive Health Library, Human Rights Watch, and Relief Web.

Data from the studies identified from the evidence reviews retrieved in stage one and the systematic literature search in stage three were extracted into a pre-piloted extraction form in Microsoft Excel developed by JM (see S3 Table). The extraction form was piloted by all authors and a meeting was held to reach consensus on the approach. We extracted information on study details (geographic area, type of study, target population, sampling), intervention details (activities, content, delivery mode, duration, mechanisms, adaptations, staff), outcomes (definitions, measures, time frame, findings) and any other notes about context and conclusions. We also included notes on how the intervention supported (or did not support) the initial programme theory model developed in stage two. Data extraction was completed by four authors (HL, DD, JD and JVM).

## Stage 4 – Purposive searches and development of final programme theories

In the final stage of this review, we used the extracted data and conducted purposive searches to refine and finalise the programme theory. Data extracted from the included studies in stage three was reviewed by three authors (HL, JD and LK) against the initial programme theory. Components of the conceptual model were removed where they were not supported by evidence and reconfigured to reflect the available evidence. For example, a group of mechanisms called 'economic empowerment' were removed from the model as there was insufficient evidence to support this intervention type from the included studies. Authors then refined the model further by grouping mechanisms into higher order themes, with a focus on how these mechanisms interact with each other and the context. For example, mechanisms relating to the design and delivery of interventions were grouped into the theme 'implementation components', and mechanisms relating to the content and its theoretical aims were grouped into themes such as 'gender transformative social norms'. The process of refinement of the model described above was used to guide the final stage of theory building in which we conducted purposive searches to link our proposed programme theories and to situate our findings across similar fields [26]. In particular, we focused on linking evidence for adolescent IPV prevention in LMICs to that of adult IPV prevention in LMICs, which is a much larger body of literature. For mechanisms pertaining to the design and delivery of interventions, we also explored the literature on effective modes of health and social intervention delivery among adolescents in different fields of research to find support for our theory.

## Results

### Interventions to prevent adolescent IPV in LMICs: Scope of the evidence base

A total of eleven adolescent IPV prevention interventions were included in this review, evaluated across fifteen individual studies (Fig 1). Nine interventions were identified from published systematic and literature reviews on the topic [9, 15–17], and two interventions were identified through database searching. Table 1 describes the characteristics of these interventions which were implemented across seven LMICs: four in South Africa, two in Mexico, and one in

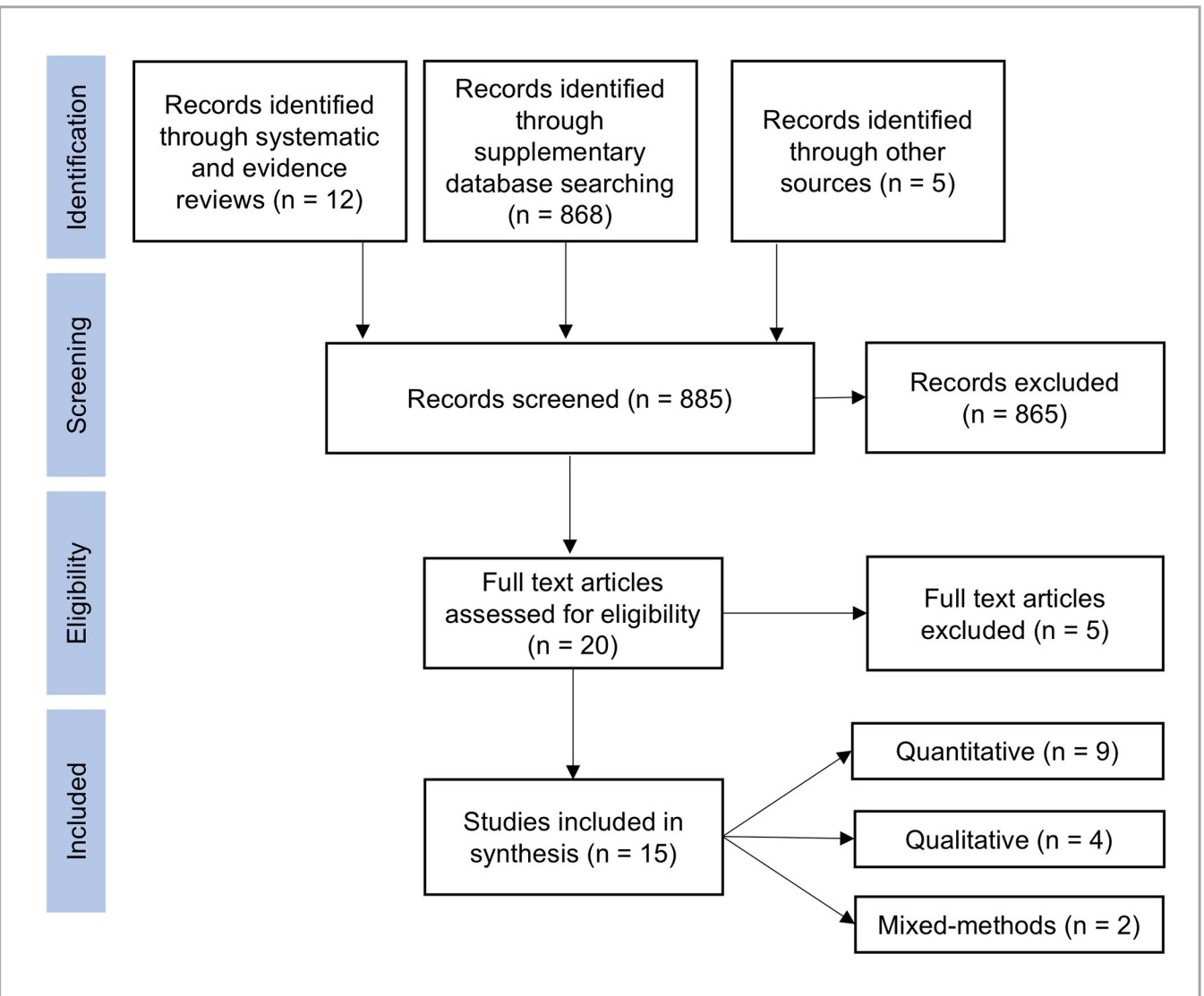

**Fig 1. Flow diagram describing number of articles selected at each stage.**

Bangladesh, Ethiopia, India, Kenya, and Malawi. The majority of interventions (n = 7) targeted both adolescent girls and boys, two targeted boys only, and two targeted girls only. Seven of the interventions were school-based and targeted adolescent boys and/or girls of high school age (generally ages 13–17 years). The other four were community-based and generally targeted adolescents and youth around 15 to 29 years. Interventions were mostly implemented in urban areas, with three including adolescents in rural areas.

Adolescent IPV was conceptualised similarly across the interventions, with most definitions mirroring those of IPV among adult populations. However, it was not clear whether interventions were focusing on IPV in relationships where both partners were adolescents or where at least one of the partners was an adolescent. To measure IPV, the majority of interventions employed the WHO Violence Against Women (VAW) survey methodology [29] focusing on physical, sexual and/or emotional IPV, while a few studies used other scales such as an adapted version of the Conflict Tactics Scale [30] or the Conflict in Adolescent Dating Relationships

**Table 1. Characteristics of included interventions.**

| Intervention | Geographic area | Target population | Setting | Intervention type | Intervention details | Study design | Sample size | Outcome |
|---|---|---|---|---|---|---|---|---|
| Sub-Saharan Africa | | | | | | | | |
| PREPARE [34] | Western Cape, South Africa | Grade 8 high school boys and girls (mean age 13.7 years) | High schools within 3-hour radius of Cape Town | Multi-component school-based HIV and IPV prevention intervention | 1. Participatory group education on gender and SRH • 21 x 1-hour weekly sessions • Single gender groups 2. School health service 3. School safety programme | Cluster randomised controlled trial (RCT) | 3,451 | Reduction in IPV victimisation among intervention arm after 12 months (OR 0.77, CI 0.61, 0.99) |
| Stepping Stones [33, 46] | Eastern Cape, South Africa | Young adult men and women aged 15–26 years (74.5% of participants were aged 15–19 years) | Rural villages | Community-based participatory education HIV prevention intervention | Participatory group education programme on gender, relationships, SRH and violence • 13 x 3-hour sessions • Single gender groups | Cluster RCT and qualitative study | 2,776 | Borderline reduction in male perpetration of physical or sexual IPV after 24 months (OR 0.62, CI 0.38, 1.01) |
| Skhokho [42] | Pretoria, South Africa | Grade 8 boys and girls aged 12–19 years | Urban high schools | School-based multi-component IPV prevention intervention | School intervention component 1. Student self-study intervention on gender, relationships and violence • 7 x 2–4 hours self-study sessions 2. Teacher training on positive discipline • 1 x 2 day training with 3 refresher sessions over 14 months 3. Break time discussion groups focused on communication, safety in relationships and coping with stress • 30 minutes during break times Family intervention component 1. Participatory group education for caregivers and their child on changes during adolescence, communication and positive discipline • Four weekend days • Peer groups of parents separate to children | Cluster RCT | 3,756 | No reduction in incidence of IPV victimisation among girls in intervention arm after 18 months (0.84, CI 0.66, 1.07) |
| Swa Koteka (HPTN 068) [44, 47] | Mpumalanga Province, South Africa | High school girls aged 13–20 years | Rural high schools | School-based conditional cash transfer intervention | Conditional cash transfer intervention • Cash transferred monthly to girls and parents based on 80% school attendance (girls: 100 Rand, parents: 200 Rand) | RCT and qualitative study | 2,433 | Reduction in experience of IPV among adolescents in treatment group after 36 months (RR 0.66, CI 0.59, 0.74) |

(*Continued*)

**Table 1.** (Continued)

| Intervention | Geographic area | Target population | Setting | Intervention type | Intervention details | Study design | Sample size | Outcome |
|---|---|---|---|---|---|---|---|---|
| Male Norms Initiative [36] | Addis Ababa, Ethiopia | Young men aged 15–24 years (58.7% of participants were aged 15–19 years) | Urban communities | Community-based multi-level IPV prevention intervention | 1. Participatory group education on gender, violence and sexual health • 8 x 2/3-hour sessions over 4 months 2. Community mobilisation activities including marches, communication materials, community dramas • 6 months | Quasi-experiment | 645 | Reduction in perpetration of physical or sexual IPV among intervention groups after 6 months (decreased from 36% to 16%, p<0.5) |
| YMOT and IMPOWER [35, 43, 45] | Nairobi, Kenya | In school boys and girls aged 15–19 years | High schools in urban informal settlements | School-based participatory education intervention for sexual violence prevention | Participatory group education with separate curriculum for boys and girls • Boys: bystander intervention, sexual consent, masculinities • Girls: empowerment and self-defence • 5 x 2-hour sessions • Single sex groups | Non-randomised longitudinal cohort and qualitative study | 489 | Reduction in the experience of sexual violence by boyfriend among intervention group after 10 months (reduction from 12.4% to 5%, p<0.004) |
|  | Lilongwe, Dedza and Salima districts, Malawi | Primary (class 5–8) and secondary (form 1–4) girls | Urban and rural districts |  | Same intervention for girls as described above, delivered across 6 x 2-hour sessions | Cluster RCT | 4,278 | No reduction in the experience of sexual violence by boyfriend among the intervention group after 10 months (increase from 44.9% to 46.4%, p = 0.46) |
| South Asia | | | | | | | | |
| SAFE [39] | Dhaka, Bangladesh | Young men and women aged 15–29 (sub-group analysis presented for ages 15–19 years) | Urban informal settlements | Multi-level IPV prevention intervention | 1. Participatory group education on gender, violence, healthy relationships • 13 x 2-hour sessions • Single sex groups 2. Community mobilisation activities including community discussions, service referral and local campaigns • 20 community groups • 1 x gender and violence training + 11 support meetings 3. Service provision • One-stop service centres providing health and legal services and referrals • Training on gender and violence for staff 4. Training and advocacy • National level advocacy on gender and VAWG • Training on gender and VAWG for civil services • Media advocacy including TV talk show on VAWG and SRHR | Three arm cluster RCT and qualitative study | 2,670 | Reduction in physical IPV towards adolescent girls aged 15–19 in the community after 18 months (RR 0.79, CI 0.62, 0.99) |

*(Continued)*

**Table 1.** (Continued)

| Intervention | Geographic area | Target population | Setting | Intervention type | Intervention details | Study design | Sample size | Outcome |
|---|---|---|---|---|---|---|---|---|
| Yaari Dosti [37] | Mumbai and Gorakhpur, India* | Young men aged 15–24 years (mean age 19 years) | Urban and rural communities | Two-component gender equality promotion intervention | 1. Participatory group education on gender, violence, SRH • 23 x 1-hour sessions over 6 months 2. Lifestyle social marketing campaign with street plays, discussions and communication materials | Quasi-experiment and qualitative study | 601 | Reduction in odds of IPV perpetration at the rural site in the intervention group after 6 months (OR 0.50, p = 0.001) |
| Central America | | | | | | | | |
| TrueLove [38] | Mexico City, Mexico | High school girls and boys (mean age 16.4 years) | Urban high schools | Multi-component school-based IPV prevention intervention | 1. Participatory group education on gender, violence, SRH, coping skills • 16 x 1-hour sessions over 16 weeks • Boys and girls together 2. School yard activities including posters, flyers, and forums 3. Training for school staff • 5 x 4-hour sessions on IPV and supporting a nonviolent atmosphere | Quasi-experiment | 1,604 | Reduction in perpetration of psychological violence among boys in intervention group by 55% after 2 weeks (p<0.05) |
| Mexfam's comprehensive sexuality education [32, 48, 49] | Mexico City, Mexico | High school girls and boys aged 14–17 years | Urban high schools | School-based participatory education IPV prevention intervention | Participatory group education on gender, violence and sexuality • 10 x 2-hour sessions • Boys and girls together | Longitudinal quasi-experiment with qualitative study (only qualitative findings published) | 157 | Intervention contributed to adolescent's development of communication skills, identification of harmful behaviours, increased help-seeking and reduced discrimination, reducing risk of IPV |

RCT: randomised controlled trial, OR: odds ratio, RR: risk ratio

*Data only included for Gorakhpur (average age in Mumbai arm was 21 years which is outside of the inclusion criteria)

Inventory [31]. Evaluation methods mostly included randomised controlled trials and quasi-experimental designs combined with qualitative studies. One study conducted a stand-alone qualitative evaluation [32]. Follow-up periods ranged from six to 36 months.

The majority of the interventions (n = 9) used participatory group-based education, including games, discussions and role plays in peer groups, for social norm change at the individual level, broadly focusing on gender, violence, relationships and sexual and reproductive health (SRH) [32–39]. These interventions will be referred to as gender transformative interventions throughout, i.e. interventions that tackle harmful gender roles, norms and power relations [40, 41]. Three of these school-based interventions included components that targeted the wider school and home environment of adolescents, for example by providing training to teachers on promoting non-violence, influencing peers outside of the intervention group with communication materials, and training parents on positive discipline [34, 38, 42]. Of the four community-based interventions, three also included community mobilisation and engagement activities to target wider communities, including activities such as social marketing campaigns

and street plays, for community level social norm change [36, 37, 39]. Only one intervention, SAFE in urban informal settlements of Dhaka, Bangladesh, attempted to foster structural change [39]. The intervention included national level media advocacy and training for civil and legal services around gender and VAW.

We only identified two interventions that did not focus on gender transformation. One school-based intervention focused solely on girls' empowerment through self-defence training for sexual assault prevention [43]. Another was a cash transfer intervention which used conditional cash transfers as an incentive to parents and girls to keep girls in school for longer, with the aim of preventing HIV and IPV [44].

**Effectiveness of adolescent IPV prevention interventions.** Nearly all the interventions showed some evidence of effectiveness in reducing the perpetration or experience of adolescent IPV, with eight presenting positive results [32, 34–39, 44] (see Table 1). Two of the school-based interventions, PREPARE in South Africa and TrueLove in Mexico City, used participatory group education alongside school-wide campaigns such as teacher training and playground activities. Both of these were effective in preventing physical, sexual or emotional IPV victimisation among boys and girls [34], and emotional IPV perpetration among boys [38]. Qualitative evaluations of the YMOT & IMPOWER intervention in Nairobi, Kenya and the Mexfam Comprehensive Sexuality (CSE) Education intervention in Mexico City, which again both used participatory group education sessions, found that the interventions reduced the risk of experiencing and perpetrating IPV for students who took part in the curriculums [32, 35]. The final school-based intervention, Skhokho in Pretoria, South Africa, which implemented a self-study curriculum, teacher training and family engagement intervention, showed no change in the incidence of IPV victimisation among girls in the intervention, which authors attribute to the study being substantially under-powered because of a smaller than anticipated sample size [42].

The two community-based interventions that targeted boys only had positive results. The Male Norms Initiative in Addis Ababa, Ethiopia, and Yaari Dosti in Mumbai, India, found that participatory group education sessions, implemented alongside broader community mobilisation activities, reduced boys self-reported perpetration of physical and/or sexual IPV [36, 37]. Despite promising results, both interventions had short follow-up periods of six months, so their sustained impact on IPV perpetration is unknown.

The remaining two community-based interventions that targeted both boys and girls also demonstrated a positive impact on IPV perpetration and victimisation. Stepping Stones in the rural Eastern Cape of South Africa, which implemented a participatory HIV and gender curriculum for boys and girls, found a borderline non-significant reduction in physical or sexual IPV perpetration among males in the intervention group after 24 months [33]. The SAFE intervention in informal settlements of Dhaka, Bangladesh found participatory group education and community-mobilisation reduced IPV experience among girls aged 15–19 in the community after 24 months [39].

Swa Koteka, the only economic intervention which used conditional cash transfers, also found a positive effect on IPV. Girls in the intervention group reported lower experiences of physical IPV up to 36 months, compared with the control group [44]. The self-defence empowerment training for girls in Kenya found a reduction in the experience of sexual assault among the intervention group, but no reductions in sexual assault by a boyfriend [45]. Given that the majority (9 out of the 11 included interventions) focus on gender transformation for adolescent IPV prevention, the next section of our results focuses on the mechanisms of change in these interventions. There was insufficient evidence to include the economic and empowerment interventions in our conceptual model.

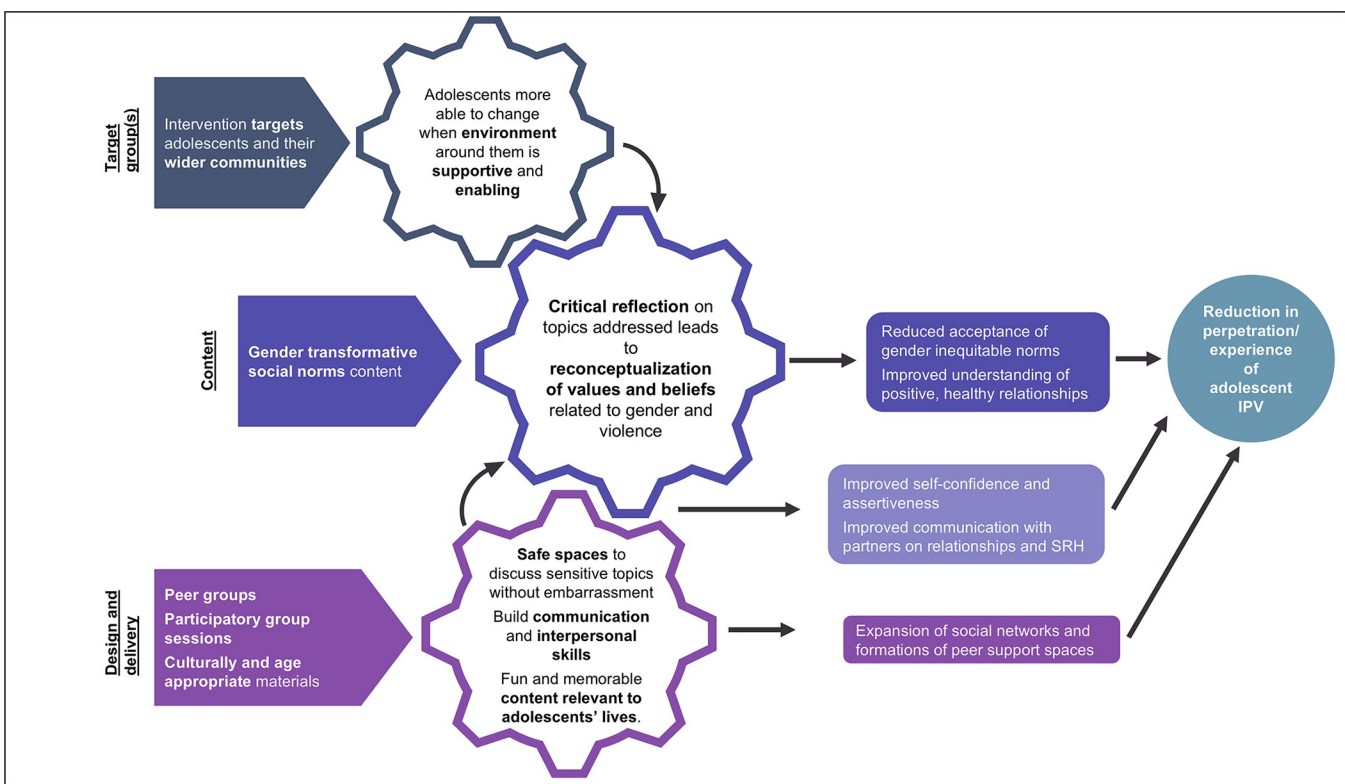

**Fig 2. Programme theory for interventions to prevent adolescent IPV.**

## Building theory for the mechanisms of change in adolescent IPV prevention

We identified three important overarching mechanisms for how gender transformative interventions work to prevent adolescent IPV (Fig 2). The central mechanism of change describes how the gender transformative content of the interventions triggered a change in reasoning among participants, resulting in overall reductions in the experience and perpetration of adolescent IPV. This central mechanism was supported by two secondary mechanisms, related to the design and delivery of the intervention and the target group. We present evidence for these mechanisms and pathways to IPV prevention under the themes of intervention theories and implementation components.

**Intervention theories: Transforming gender attitudes and norms for behaviour change.**    All but two of the interventions included in this review focused on transforming attitudes and norms around gender and violence for the prevention of IPV among adolescents. Gender transformative interventions included content focused on gender, violence, power, relationships and SRH, usually in the form of participatory group education, in either schools or communities. Focusing on gendered social norms and attitudes in these supportive peer spaces triggered adolescents to question the social norms in their lives that perpetuate gender inequality and drive IPV [32, 37, 39]. The content also prompted adolescents to critically reflect on their own relationships and identify their own and their partner's attitudes and behaviours that are harmful and need to change [32, 33, 37, 39]. There was clear support for this mechanism in a longitudinal qualitative analysis of Mexfam's school-based comprehensive sexuality education intervention in Mexico City. Participants engaged in self-reflection in light of the content presented during the course, particularly around concepts of jealously and

controlling behaviour [48]. The reconceptualisation of these concepts as types of violence, rather than expressions of love, led to participants stating new intentions for their relationships, and presenting more positive relationship narratives in later interviews. The mechanism of critical reflection on gender attitudes, norms and power imbalances is noted elsewhere as a key component of successful IPV prevention interventions with adult men and women [50–52].

Interventions with boys that also focused on gender transformation, but from a masculinity perspective, found that boys-only groups also undergo a similar process of reconceptualising their beliefs on gender, relationships and violence. Content in both interventions, generally focused on how masculinities can be harmful, and what positive masculinities can look like, prompted boys to reflect on what it means to be a man in their society, and how gendered social norms have shaped their perceptions [36, 37]. It provided a safe space for boys to explore these deeply personal concepts with similar peers, and the opportunity to reject harmful masculinities and build positive concepts around how they would like to be a man in the future. These types of interventions, which have been described as 'transforming masculinities' [53], are emerging as a key focus of many violence prevention interventions with men, particularly in LMICs, and have shown promising results for social norm and attitude change around violence, fatherhood, and IPV reduction [54–56].

This process of reconceptualising beliefs about gender, relationships and violence was not identified as a mechanism by authors across all gender transformative intervention evaluations, however, a large number of evaluations reported a reduced acceptance of gender inequitable attitudes or norms as an intermediary outcome [32, 33, 35–39, 42]. Seven of these eight interventions also found significant reductions in the perpetration or experience of adolescent IPV, suggesting that changing norms around gender and violence might have been effective in reducing IPV. In additional analyses of two of these interventions, it was found that being more supportive of gender equitable norms was associated with a lower perpetration of physical or sexual violence at endline among boys [36, 37]. These findings are supported by evidence of how interventions work to prevent IPV in adult populations, with a number of large-scale interventions finding similar trends in reductions in norms predicting reductions in violent acts, such as in the SASA! Community Mobilisation trial in Uganda [57]. There are however longstanding debates as to whether changing attitudes and norms actually leads to behaviour change [58].

The unique developmental and social context of adolescence may also be important in understanding why interventions were successful in triggering adolescents to reconceptualise their ideas and beliefs around gender and relationships. The SAFE intervention in informal settlements of Dhaka found the intervention was more effective for younger groups. Reductions in IPV experience were found among the 15–19 year old subgroup, but not among the wider intervention target group (15–29 years). In an unpublished qualitative evaluation (retrieved through correspondence with the study team), authors suggest this may be attributable to them having experienced fewer relationships resulting in them being less exposed to violence and having had less opportunity for negative behaviours and beliefs to become normalised. While there was little evidence to support this mechanism across the other evaluations, targeting adolescents for social norm change on other health and social issues such as teenage pregnancy and marriage have found similar results: more positive attitudinal change among adolescents aged 10–14 years compared to adult members of their family [59]. The What Works to Prevent VAWG initiative also found trends towards greater intervention impacts on some types of IPV among adolescents and women under 25, compared to older women [60].

The context seemed to play an important role in determining how successful gender transformative interventions were, and which sub-groups they were effective for. Despite targeting

both boys and girls using the same curriculum, True Love in Mexico City only found a reduction in the perpetration of IPV among boys, not experience among girls [38]. Similarly, Stepping Stones in South Africa also only demonstrated a borderline reduction among boys in the intervention [33]. In their qualitative evaluation, authors argued that in this cultural context, in which a young woman's status is intrinsically linked to her being in a relationship, it may have been too risky to change her beliefs and behaviour. By carving out more empowered feminist identities for herself based on the content learned during the intervention, a young woman risked being rejected by her boyfriend, which could have implications for her social status. Some gender transformative IPV prevention interventions with adults in LMICs have also found the interventions only reduced IPV perpetration among men, and not experience among women [61].

Mexfam's comprehensive sexuality education intervention was the only intervention to report addressing sexual diversity and tackling homophobic discrimination in their gender transformative intervention [32]. The study team reflected on how lesbian, gay, bisexual and questioning participants in particular reported benefiting greatly from the intervention, which they believed may be because these groups of students might have been particularly motivated to engage if they were grappling with their own identity.

This interaction of the intervention mechanisms with the context is also supported by evidence from the SAFE intervention in Dhaka, in which there were only positive results when both boys and girls were targeted, rather than just girls alone [39]. Girls may be more able to hold more gender equitable beliefs when the boys around them are also supportive of this. Similarly, targeting wider communities, as well as adolescents, for social norm change around gender and violence seemed to contribute to the effectiveness of gender transformation interventions [36, 37, 39]. There was little exploration of why this was the case, despite the mechanism of wider community norm transformation supporting individual level behaviour change in the broader literature for community-based IPV prevention [57]. Including community-level components may also have been effective in reducing IPV because it provided adolescents with opportunities to engage in activism and become agents of change in their communities by sharing their learnings and challenging beliefs and behaviours of those around them [32, 39]. Community activism is recognised more broadly as a key strategy for the prevention of VAW [62].

The school-based gender transformative interventions which included parental and teacher training components, while demonstrating some promising results, provided limited evidence in support of how broader school and home engagement can lead to adolescent IPV prevention. Positive changes were seen in relation to the parental component of the Skhokho intervention (improved communication with children, improved gender attitudes and reductions in reported childhood trauma), but there was no control arm so the relative impact of this component is not well understood [42]. Similarly, interventions that included teacher training and engagement fell short in exploring how these components may have contributed to IPV prevention [34, 38].

**Implementation components: Contextually relevant, participatory peer-group education.** How interventions were designed and delivered was important in supporting adolescents to undergo the process of reconceptualising their values and beliefs related to gender and violence, the mechanism central to intervention success. The mechanisms triggered by these implementation components are depicted as supporting the central mechanism in Fig 2.

The majority of interventions included in this review used participatory group learning sessions as the principal mode of delivery. Group sessions included learning, discussing, debating, role playing, and games, amongst other interactive and engaging methods. Despite intervention successes being attributed to this delivery method, there was little information from

authors on the mechanisms behind the success of participatory group education with adolescents for IPV prevention. Broader literature on IPV prevention with adults and children does however support this mechanism, finding that participatory approaches promote experiential learning (learning by doing), and are critical to the success of educational interventions to prevent VAW [63].

Culturally and age-appropriate content also provided a mechanism that supported adolescents on the pathway to reconceptualization and social norm and behaviour change. Most of the included interventions delivered programmes with materials and content adapted to the specific context and participant age group, for example by ensuring characters and storylines were locally relevant and focused on issues affecting the age group they were targeting, and by using age-appropriate methods such as games and role plays to keep participants engaged and focused [32–35, 37, 39, 42]. Two of the interventions used materials that were co-developed by intervention participants themselves [33, 42]. When adolescents perceive the content to be relevant and resonate with their own lives, they engage more, which sparks greater self-reflection and participant buy-in [33, 35, 39]. This mechanism and the need for culturally and age-appropriate materials for intervention success is widely supported by VAW prevention interventions with adults [63].

Most of the interventions were delivered in single gender peer-groups [33–37, 39, 42]. This implementation style also provided a mechanism in support of greater critical reflection because adolescents perceived the groups as safe spaces where they could engage, reflect and discuss in a supportive environment without embarrassment or ridicule [32, 35, 39]. In the YMOT & IMPOWER intervention in Nairobi, peer-groups were spaces where boys could express themselves freely, which was a space that had not existed for them before the intervention [35]. Similarly, having peer facilitators who were skilled in navigating sensitive discussions [32, 33, 35–37], and who adolescents perceived as relatable and similar to themselves, further promoted an environment that was conducive to engagement and critical reflection [32, 35, 36, 39]. Other research highlights the potential of peer-facilitated community-based interventions with adolescents in LMICs for improving mental health and reducing substance use [64].

The interactive, group-based nature of most of the interventions contributed to adolescents developing communication and interpersonal skills [32, 39, 42], building self-confidence [32, 33, 35, 39] and expanding their social networks [32, 39], all of which we hypothesise to contribute to reductions in the perpetration and experience of IPV among adolescents (Fig 2). Firstly, discussing and debating sensitive issues in safe group spaces developed adolescents' communication and interpersonal skills, which in turn had implications for their relationships. They were able to communicate more assertively and effectively with their intimate partners and had improved conflict resolution skills [32, 37, 39, 46], which were hypothesised to have improved relationship quality and reduced IPV in other interventions with adult populations [65]. While there was less evidence for how building relationships with peers and expanding social networks contributed to reducing IPV in the studies included in this review, evidence from interventions in other fields of research shows that social cohesion developed from peer-group interventions can be leveraged to achieve intended outcomes through mechanisms such as the provision of mutual support [66].

When adolescents were able to reflect and build positive concepts of gender and relationships in a safe and supportive environment, supported by wider awareness and change at the community level, interventions were more successful in reducing adolescent IPV.

## Discussion

This realist review set out to identify the mechanisms that bring about change to prevent adolescent IPV in primary prevention interventions in LMICs. We identified three groups of

mechanisms related to the specific intervention content, the design and delivery of interventions, and the intervention target groups. The triggering of these mechanisms worked synergistically to bring about the intended change: reductions in the perpetration and/or experience of adolescent IPV.

The majority of interventions used school-based or community-based interactive peer-group education to transform attitudes and norms around gender and relationships. The gender transformative content presented to adolescents prompted them to critically reflect on their attitudes towards gender and violence and their behaviours in their own intimate/romantic relationships, guiding them towards a process of reconceptualising what constitutes violence, and what is acceptable behaviour. This process of change was supported by peer-group based interactive interventions being safe spaces for open discussion and opportunities for skill building, as well as supportive environments at the community level.

Promoting and fostering more gender equitable attitudes and norms for behaviour change is not unique to adolescent IPV prevention, and gender transformation has been a central focus of many VAW prevention interventions in the last two decades [67]. Interventions are "gender transformative" when they actively seek to reshape gender relations to be more equitable for the improvement of human health [40, 68]. This approach implicitly assumes that reductions in support for the social norms and attitudes that legitimate and perpetuate VAW will indeed lead to a reduction in perpetration and experience [69, 70]. This pathway of norm/attitude change leading to behaviour change can be traced back to the work of social psychologists who argued that behaviours are predicted by behavioural intentions, which are in turn a function of individuals' attitudes and social norms [58, 71, 72]. This pathway however is often contested, and in reality there is insufficient empirical evidence to support this direction of influence. It is likely that the relationship between attitudes/norms and behaviour is more bi-directional, and that behaviour is also influenced by many other contextual factors [73, 74]. As such, it is difficult to determine the exact pathway through which gender transformative interventions, such as those included in this review, achieve change. Nevertheless, a growing number of interventions, including some in this review, have been able to show quantitatively the relationship between increasing support for gender equitable attitudes and norms and reduced VAW [36, 37]. Similarly, the qualitative studies in this review provide evidence in support of this pathway [35, 46, 48], particularly the longitudinal repeat qualitative data from Mexfam's comprehensive sexuality education intervention, which was able to show incremental changes in participants attitudes towards different types of IPV over time and discussions in later interviews of how they had changed harmful behaviours in their relationships as a result [48].

Another barrier to understanding how gender transformative interventions achieve change is the fact that despite focusing on shifting gendered social norms, many of these interventions only tend to measure individual attitudes and not social norms themselves [41, 75]. The majority of interventions included in this review took a norms approach to adolescent IPV prevention without explicitly testing the assumptions of social norms theory and measuring changes in social norms [76]. This is a critique of many gender transformative interventions and drives the need for better tools to measure social norms for understanding the long term impact and sustainability of gender transformative interventions [40]. Such interventions have also been critiqued for a narrow and heteronormative focus on gender, without fully considering the intersection of gender with other social structures that might drive VAW, including sexuality, class and racial inequalities [40, 77]. Evidence for IPV prevention among LGBTQI+ groups has been identified as a gap in the evidence for IPV prevention, despite the disproportionately high rates of violence they experience [67]. Proponents argue that for gender transformative interventions to be truly transformational, they must address the multiple intersecting identities of participants. An intersectional approach will contribute to building the evidence around

which sub-group groups these interventions are most effective for, which is critical when considering who gender transformative interventions should target and what formative research is required to understand the lives and lived realities of these groups in a particular context.

Understanding why some gender transformative IPV prevention interventions only appear to be effective among boys and men is also important when considering the implementation of these interventions in different contexts. One explanation relates to the interaction of the intervention mechanisms with the context. For example, in an evaluation of a later iteration of the Stepping Stones intervention in South Africa, with predominantly young adult men and women, some men were able to change their behaviour because there were acceptable alternatives to the dominant masculinity [78], unlike for women in the earlier Stepping Stones intervention for whom change was too risky [46]. Instead of completely reconstructing a new masculinity, men could make subtle changes towards less violent identities while still meeting masculine ideals in the wider social context, such as being a provider for their family. While we cannot rule out the possibility of social desirability bias among intervention men in response to the post-intervention surveys as an alternative explanation, a thorough understanding of the context and local dynamics can contribute to understanding why these interventions may be more successful among certain groups. This also highlights the potential of realist evaluation methods for exploring how the context influences intervention success for different groups and how interventions might be adapted to account for this.

In this review, all of the gender transformative interventions with adolescents demonstrated some positive impact on the experience and/or perpetration of adolescent IPV. The unique transitional period of adolescence might be helpful in understanding these successes, and why other gender transformative interventions have seen trends towards greater reductions in IPV among adolescents and women under 25, compared to adult women (specifically for sexual and economic IPV) [60]. While gender socialisation begins from birth, there is ample evidence to suggest that adolescence is a critical period for the shaping and solidifying of gender expectations, attitudes and related behaviours, which are also heavily influenced by parents, peers and broader social networks during this period [10, 79, 80]. Subsequently, interventions that aim to construct more positive gender expectations, attitudes and social norms among adolescents may be particularly salient in this group who are still in the process of establishing their beliefs, and who have not yet fully normalised gender inequitable attitudes and behaviours [12, 79, 81]. This is not to say that gender transformative interventions are only effective with adolescents and younger adults, but further highlights the potential for intervening during this critical transitional period to set the foundations for healthy future relationships. The majority of interventions in this review targeted older adolescents, aged 15–19 years, despite much evidence suggesting that the period of younger adolescence, 10–14 years, could be even more critical for establishing gender equitable attitudes and social norms [12]. Of the interventions that did also target younger adolescents, none included analyses of intervention success among younger compared to older adolescents. As such, an avenue for future research would be to evaluate exposure to similar gender transformative interventions among younger adolescents, to fully understand the true transformative potential of these interventions across the huge developmental period of adolescence.

Adolescents' gender attitudes are largely influenced by their peers, parents and wider social circles, so adolescent IPV prevention interventions should move beyond an individual level focus and target wider social networks of adolescents [10]. Around half of the interventions in this review included components that targeted the wider school, home and community environments of adolescents, but theorisations of how these components contributed to and supported gender transformation and IPV prevention at the individual level was not well explored. More work is needed to understand how the various strategies included in multi-

component interventions can complement each other and work synergistically to achieve the shared outcome. Only one intervention built on community engagement activities to attempt to foster change at the structural level [39]. Despite efforts to target gender transformation at multiple levels of the ecological framework [82], interventions focusing on gendered social norms at the individual, interpersonal and community levels have been critiqued for assuming that behaviours are completely agentic choices and are modifiable without broader change at the structural level [40]. As Dworkin describes for gender-transformative health programming with men, individual and group-based behaviours must be viewed within the social, economic and cultural contexts that shape them, including consideration of key structural determinants such as poverty, migration, conflict, globalisation and racism [40]. Evidence on structural interventions, which aim to modify the aspects of the economic, politico-legal, physical and social environment that produce and reproduce IPV risk [83], is also limited for IPV prevention among adult populations and is an important avenue for future research and evaluation [84].

Similarly, there is a skew towards focusing on transforming gendered social norms and attitudes for adolescent IPV prevention, when norms and attitudes are only one of many drivers of IPV [1, 85, 86]. We only identified one intervention that had a central focus on an important and broadly recognised protective factor for IPV: education [1, 87]. Providing girls and their families with cash payments for school attendance significantly reduced girls' experience of IPV in rural South Africa in the SwaKoteka intervention [44], highlighting the potential of mechanisms beyond gender transformation in the prevention of adolescent IPV, such as increasing agency and autonomy among girls. There is strong evidence for cash transfers improving school enrolment and similar economic interventions have also demonstrated promising results for preventing HIV among adolescent girls [88]. The girls component of the YMOT & IMPOWER intervention in Nairobi, Kenya, which focused on empowerment and self-defence, provides some support for mechanisms of increased agency and assertiveness as contributing to reduced sexual assault by a boyfriend [35]. However, this finding did not hold when the same intervention was implemented in Malawi [43]. While sexual violence by any perpetrator reduced for girls in the intervention group, there was no change in perpetration by boyfriends, meaning we could not identify strong support for these alternative mechanisms of empowerment and self-defence. Interventions targeting other important drivers of adolescent IPV, including adverse childhood experiences, were also largely missing from the literature on IPV prevention with adolescents, despite a large body of literature highlighting their importance for preventing the upstream drivers of adolescent IPV [9, 89]. The Skhokho intervention in South Africa was an exception to this, which included a positive parenting component to promote non-violent discipline at home alongside a school-based education programme for transforming gendered social norms [42]. However, this component lacked a control arm and so its individual impact on adolescent IPV is unknown.

The realist approach is a major strength of this review. There are a growing number of interventions targeting adolescent IPV prevention in LMICs, yet very few evaluation studies explore the mechanisms through which these interventions achieve change. This review uses realist evaluation to build a theoretically grounded model of the mechanisms of change in these interventions, to understand how they work, for whom and under which circumstances, which makes a valuable contribution to the evidence base. There are several limitations to this review. Firstly, realist syntheses do not employ exhaustive database searching strategies as are used in systematic reviews, which means our list of included interventions may not represent every published evaluation that meets the inclusion criteria. However, we drew on recently published evidence reviews and conducted additional database searching to retrieve relevant studies. Secondly, only including interventions that explored behavioural outcomes related to

adolescent IPV (perpetration or experience) may have prevented us from including insights from similar interventions that only assessed changes in knowledge and attitudes as their outcomes of interest. These studies, while not able to provide evidence for the impact of the intervention on behaviour, may have included relevant insights for understanding mechanisms of change. The final limitation relates to the lack of qualitative and longitudinal intervention evaluations. Some quantitative evaluations of quasi-experiments and RCTs lacked theorisations about the mechanisms through which the interventions achieved change. As such, analysing the data presented in these studies using a realist framework was challenging and we often had to draw upon wider literature on similar interventions in different populations to explore the potential mechanisms. Finally, the short-term follow-up periods of interventions limited our understanding of their sustainability. Nevertheless, the realist approach is flexible and iterative and enabled us to draw on broader literature to inform our conclusions about these pathways of change.

## Conclusion

Gender transformative interventions to prevent IPV among adolescents work by engaging girls and boys in critical reflection on their attitudes and behaviours related to gender, violence and relationships, in safe and supportive environments. They also promote self-confidence, communication skills, and the expansion of peer networks, which have benefits beyond IPV prevention. This review highlights the immense potential of gender transformative interventions during the adolescent years in which effective interventions can lay the foundations for future relationships free from violence. To maximise their potential, gender transformative interventions should include both boys and girls, target early and later adolescence, and the wider social and community networks of adolescents. However, the focus on gender transformation must not obscure attention from the importance of intersectionality informed approaches and other important yet understudied drivers of adolescent IPV. IPV during adolescence must be addressed holistically by understanding how broader contexts of adverse childhood experiences, racial and economic inequalities, and conflict and displacement also pattern together with gender inequality and social norms to create high risk environments for adolescent IPV. Finally, future interventions should consider expanding their methodological approach to include longitudinal methods such as repeat qualitative interviews to enable a more fine-tuned understanding of the incremental and non-linear pathways in which multi-component and multi-level IPV prevention interventions achieve change for adolescents.

## Supporting information

**S1 Table. Search strategy.**
(DOCX)

**S2 Table. Realist synthesis reporting standards.**
(DOCX)

**S3 Table. Data extraction form.**
(DOCX)

**S4 Table. Dataset.**
(DOCX)

## Author Contributions

**Conceptualization:** Hattie Lowe, Joanna Dobbin, Ligia Kiss, Joelle Mak, Jenevieve Mannell, Daniella Watson, Delanjathan Devakumar.

**Formal analysis:** Hattie Lowe, Joanna Dobbin, Jenevieve Mannell, Delanjathan Devakumar.

**Methodology:** Hattie Lowe, Ligia Kiss, Joelle Mak, Delanjathan Devakumar.

**Project administration:** Delanjathan Devakumar.

**Supervision:** Ligia Kiss, Delanjathan Devakumar.

**Writing – original draft:** Hattie Lowe.

**Writing – review & editing:** Hattie Lowe, Joanna Dobbin, Ligia Kiss, Joelle Mak, Jenevieve Mannell, Daniella Watson, Delanjathan Devakumar.

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
