## [Decision Letter · Decision Letter 0]

16 Aug 2022

PGPH-D-22-01065

Mechanisms for the prevention of adolescent intimate partner violence: a realist review of interventions in low- and middle-income countries

Dear Dr. Lowe,

Thank you for submitting your manuscript to PLOS Global Public Health. After careful consideration, we feel that it has merit but does not fully meet PLOS Global Public Health’s publication criteria as it currently stands. Therefore, we invite you to submit a revised version of the manuscript that addresses the points raised during the review process.

We look forward to receiving your revised manuscript.

Kind regards,

Rakesh Singh

Academic Editor

Journal Requirements:

Review Comments to the Author

Reviewer #1: Thank you for asking me to review this manuscript. This is a an overall well written manuscript describing a well conducted realist review of interventions that seek to prevent intimate partner violence in adolescents, focusing on low- an middle income countries. The contribution of the manuscript in generating knowledge about the mechanisms for interventions is important for those who seek to fund, develop, implement and evaluate interventions to reduce IPV, which a major public health concern. There are several limitations of the review e.g. with regards to its scope, which are well explained.

The following minor revisions are needed:

- a careful spell check throughout manuscript is required e.g. RAMSES instead of RAMESES

- the authors are not including adverse effects in their programme theory - considering the nature of the topic it would be good to explain how this was or was not investigated, and why

- whilst it is fully understood that the realist review method is complex and it is difficult to be fully transparent about how the the programme theory was derived from the data, it would be helpful if authors could share the full data extraction tables (authors mention they used a Microsoft Excel document and it might be helpful to share this)

- whilst authors explain some of the limitations of the review method, there is not much information about limitations with regards to applying the realist review method to developing the programme theory; in particular what were challenges around including findings of studies that did not provide sufficient detail for the purpose of informing a programme theory; was the relevance of studies assessed?

Reviewer #2: There is a minor linguistic edit required to the piece. Additionally the piece would benefit more from a clearer methods/analysis section. Though the data is adequately presented, the paper lacks a bit of a walkthrough to how these results were achieved.

The paper would also benefit from a stronger wrap-up section, bringing the discussion back to why IPV is in fact an 'adolescent' issue through an intersectional lens. Why it is pivotal for IPV among adolescents to be separated from larger work on GBV and IPV among adults, and how impeding factors such as conflict, poverty, displacement and social norms frame this. A more elaborate and comprehensive conclusion will lay the foundation for further research, and adequately frame findings within the necessary intersectional realities.

Reviewer #3: Thank you for producing an interesting analysis of adolescent intimate partner violence and the mechanisms for preventing it. I have some feedback/concerns in this paper before publishing it.

Page 13; Para 2: The authors argue that follow-up after 6 months might not be appropriate for observing the sustainability of impact. What can be the minimal duration of follow-up for observing the impact?

Page 21-27: The findings and discussion might have intermixed here. Some content from this section can be moved to discussion section.

Page 26 para 1 and Figure 2: The authors have identified peer pressure was effective in reduction in perpetration and experience of IPV among adolescents. However, several meta-analysis in the past have reported that peer education have limited effects in promoting healthy behaviors and improving health outcomes among target groups. Those evidence have reported that peer education programs mainly benefit peer educators rather than their intended beneficiaries. Can the authors discuss about this ? The authors can go through https://www.ghspjournal.org/content/ghsp/3/3/333.full.pdf for further details.

Minor

Page 11, Line 216: The countries should be 7. Please explain if any of this is not LMIC.

Page 11: Line 235: ‘Nearly all’ does not add the value here.

Page 11, Line 237: Reference Typo (32-39)??

Page 13: Line 290: Reduction in experience of sexual assault among the intervention group after self-defense training for girls was from Kenya while Malawi study did not show any reduction in sexual assault by boyfriends as per Table -1: Please check it.

Page 23, Para 1: The definition of Adolescents might be appropriate in Background section rather than in the findings section. Some sentences might require references here.

---

## [Decision Letter · Decision Letter 1]

11 Oct 2022

Mechanisms for the prevention of adolescent intimate partner violence: a realist review of interventions in low- and middle-income countries

PGPH-D-22-01065R1

Dear Miss Lowe,

We are pleased to inform you that your manuscript 'Mechanisms for the prevention of adolescent intimate partner violence: a realist review of interventions in low- and middle-income countries' has been provisionally accepted for publication in PLOS Global Public Health.

Best regards,

Rakesh Singh

Academic Editor

Reviewer Comments:

Reviewer #1: All comments have been addressed

Reviewer #2: All comments have been addressed

Reviewer #3: All comments have been addressed